# The Effects of Volume-Matched One-Day Versus Two-Day Eccentric Training on Physical Performance in Male Youth Soccer Players

**DOI:** 10.3390/jfmk10030260

**Published:** 2025-07-09

**Authors:** Raja Bouguezzi, Yassine Negra, Senda Sammoud, Helmi Chaabene

**Affiliations:** 1Research Laboratory (LR23JS01) “Sport Performance, Health & Society”, Manouba 2010, Tunisia; rajabouguezzi@hotmail.com (R.B.); yassinenegra@hotmail.fr (Y.N.); senda.sammoud@gmail.com (S.S.); 2Higher Institute of Sport and Physical Education of Ksar Saïd, University of “La Manouba”, Manouba 2037, Tunisia; 3Institut Supérieur de Sport et de l’Éducation Physique du Kef, Université de Jendouba, Le Kef 7100, Tunisia; 4University Hospital Magdeburg, Division of Cardiology and Angiology, Otto-von-Guericke University Magdeburg, 39106 Magdeburg, Germany

**Keywords:** youth athletes, team sports, training frequency, physical fitness, muscle lengthening

## Abstract

**Objectives**: This randomized controlled trial investigated the effects of an 8-week, volume-equated eccentric training program comprising Nordic hamstring and reverse Nordic exercises performed either once or twice per week on measures of physical fitness in pubertal male soccer players. **Methods:** A total of 34 participants were randomly assigned into a 1-day (*n* = 16; age = 14.58 ± 0.28 years) or 2-day (*n* = 18; age = 14.84 ± 0.22 years) per week training group. Physical fitness was assessed using 5 m and 10 m sprints, the 505 change in direction (CiD) speed test, Y-shaped agility test, countermovement jump (CMJ), and standing long jump (SLJ). **Results**: Significant group-by-time interactions were observed for the 505 CiD test, agility, and CMJ performance (effect sizes [ES] = 0.80 to 1.13). However, no significant interactions were found for the 5 and 10 m sprints or for SLJ (*p* > 0.05). Compared to the 1-day group, the 2-day training group showed greater improvements in CiD speed (∆7.36%; *p* < 0.001; ES = 0.92), agility (∆7.91%; *p* < 0.001; ES = 1.68), and CMJ (∆7.44%; *p* < 0.01; ES = 0.35), while no differences were observed in 5 and 10 m linear sprints or SLJ performance. According to individual response analysis, improvements across the physical fitness parameters beyond the smallest worthwhile change (SWC_0.2_) were observed in 22–83% of players in the 1-day group and 77–100% in the 2-day group. **Conclusions**: In summary, the findings suggest that when training volume is matched, distributing the eccentric training regimen over two days per week may lead to greater improvements in CiD speed, agility, and CMJ performance compared to a single-day approach.

## 1. Introduction

Resistance training is now a fundamental component of athletic conditioning [1,2,3], including for youth athletes [4,5]. More specifically, eccentric training has gained substantial popularity in recent years [6,7]. An increasing body of evidence suggests that resistance training programs that adequately load the eccentric phase of movement can induce superior neuromuscular adaptations compared to concentric-only or traditional resistance training (i.e., combined eccentric and concentric muscle actions) [8,9,10]. Additionally, strong evidence indicates that eccentric training results in higher muscle strength, muscle power, and hypertrophy compared to concentric-only or traditional resistance training [6,7,8,9,10,11,12]. Eccentric training has recently been advocated for youth athletes [13]. Specifically, Bright et al. [13] conducted a systematic review and concluded that existing evidence supports incorporating eccentric training to enhance muscular strength, jumping ability, sprint performance, and change in direction (CiD) in youth athletes. However, the authors noted that eccentric training methods are largely limited to the Nordic hamstring exercise (NHE) and flywheel inertial training.

Clearly, eccentric training is a valuable component of athletic preparation in youth, especially in sports requiring high-intensity actions and rapid CiD, such as soccer. Despite its potential benefits, research on eccentric training in youth athletes, including youth soccer players, remains limited [13]. Most eccentric training studies have focused on young adults [6]. Practically speaking, establishing effective eccentric exercises that are easy to implement is crucial. In this context, the NHE (e.g., individuals positioned on their knees using their hamstring muscles to resist a forward-falling motion) is easy to integrate as well as effective in improving a wide range of physical fitness measures [4,14,15]. Recently, the reverse Nordic exercise (e.g., individuals positioned on their knees resisting a backward-falling motion [RNE]) has shown promising results with positive effects on key measures of physical fitness such as sprinting, jumping, and CiD in youth [16]. Additionally, there is evidence that 8 weeks, with two sessions per week, of RNE training resulted in a large increase in muscle fascicle length, muscle thickness, pennation angle, and cross-sectional area in young adults aged 24 years [17]. However, to the best of the authors’ knowledge, despite the promising empirical evidence supporting the individual benefits of the NHE and RNE on key physical components essential for youth soccer players, such as strength, power, and CiD, there is no research specifically focusing on the combined effects of the NHE and RNE within the same training program.

Moreover, limited time is often identified as a significant barrier to youth participation in sports [18]. Time restrictions have been identified as a frequent factor preventing coaches from implementing resistance training with youth athletes [19]. Given the crucial role of eccentric training in promoting physical fitness in youth [13,20,21], it is essential for coaches to effectively incorporate time-efficient methods into their weekly routines. While the ideal frequency for eccentric training remains unclear, studies by Abdelkader et al. [22] and Moran et al. [15] suggest that NHE training at 1 or 2 sessions per week can improve high-intensity athletic tasks and physical fitness, with slight benefits from spreading sessions across two days. However, the effects of training frequency in combined NHE and RNE have not been investigated yet, pointing to a void in the literature.

Considering the evidence presented above, we sought to compare the impact of volume-equated 1-day- and 2-day-per-week combined NHE and RNE on measures of physical fitness in pubertal male soccer players. Building upon previous findings, we hypothesized that both training interventions generate beneficial effects and that spreading training over 2 days has an advantage over 1 day [15,22].

## 2. Methods

### 2.1. Experimental Approach to the Problem

A randomized controlled trial was undertaken to compare the effects of an 8-week, 1-session vs. 2-session weekly, volume-matched eccentric training on measures of linear sprint, CiD speed, agility, and jumping performance in youth male soccer players. The training programs were implemented during the in-season period of 2024 (November–December). All tests were scheduled at least 48 h after the last conducted training session or soccer match and were carried out at the same time of day (7:30–9:30 AM). Testing took place over two sessions with linear sprint speed, CiD speed, and agility testing conducted on the first day, and jump height testing on the second day.

### 2.2. Participants

We conducted a priori sample size calculation using G*Power software (version 3.1.6). We set a type I error rate of 0.05 and 80% statistical power. The estimated effect size of Cohen’s d = 1.18 for the outcome 505 CiD speed was based on a similar study from Bouguezzi et al. [16] on the effects of the RNE exercise on physical fitness in youth karate athletes. The analysis indicated that a total of 20 participants would represent a sufficient sample. To account for potential attrition, 34 participants from a regional soccer team were recruited. Participants were randomly assigned to a 2-day group (*n* = 18) or a 1-day group (*n* = 16), using sealed, opaque envelopes to ensure allocation concealment. The anthropometric characteristics of both groups are displayed in Table 1. The participants had 4.0 ± 2.1 years of systematic soccer training and competition, involving three to four training sessions (80–90 min each) per week and a competitive game on weekends. Players who missed more than 20% of the total training sessions and/or more than two consecutive sessions were excluded from the study. Biological maturity status was estimated using the maturity offset (MO) method. The MO was calculated by predicting age at peak height velocity using the estimation equation established by Moore et al. [23].MO = −7.999994 + (0.0036124 × age × height)

All procedures were approved by the Institutional Review Committee for the ethical use of human subjects at the Higher Institute of Sport and Physical Education, Ksar Said, Tunisia (LR24JS15). Written informed parental consent and participant assent were obtained before the start of the study. All participants and their parents/legal representatives were informed about the experimental protocol and its potential risks and benefits before the commencement of the research project. Participants were permitted to withdraw from the study at any time and without having to provide a reason for doing so.

### 2.3. Linear Sprint Speed Time

Ten-meter linear sprint performance, including a 5-m split time, was assessed using an electronic timing system (wittygate, Microgate, SRL, Bolzano, Italy). Participants started in a standing split stance position with their lead foot 0.3 m behind the first infrared photoelectric gate, which was placed 0.75 m above the ground to ensure that it captured trunk movement and avoided false signals through limb motion. In total, three single-beam photoelectric gates were used. No rocking or false steps were permitted before starting. The between-trial recovery time was three minutes. The best performance out of two trials was used for further analysis.

### 2.4. The 505 Change in Direction

The 505 CiD speed test was administered according to Negra et al. [24], with performance assessed by using an electronic timing system (Wittygate, Microgate, SRL, Bolzano, Italy). From a standing position at 10 m from the start line, players ran as quickly as possible through the start/finish line, pivoted 180° at the 15 m line indicated by a cone marker, and returned as fast as possible through the start/finish line. To ensure proper execution of the test, a researcher was positioned at the turning line and if the participant changed direction before reaching the turning point, the trial was disregarded and reattempted after a three-minute recovery period. A between-trial rest period of three minutes was provided. The best performance out of two trials was used for further analysis.

### 2.5. Y-Agility Shaped Test

The Y-shaped agility test was administered as per Lockie et al. [25]. The Witty light-based timing system was used to record the time and set the reactive conditions. The width of the gates was 1.5 m with a height of 1.2 m. From 0.3 m behind the start line, participants performed a 5 m maximal straight sprint. Then, they performed the CiD task as quickly as possible with a 45-degree CiD to the left or the right side, followed by a 5 m-long sprint through the gates. To dictate the direction, a green arrow appeared with a delay of approximately 40–45 ms after passing the starting gate. Two trials were performed, and the best time was taken for further analysis. A rest period of 90 s was allowed between trials.

### 2.6. Countermovement Jump

From a standing position with feet shoulder-width apart and arms akimbo, participants performed a fast downward movement before performing a maximal vertical jump. Countermovement jump (CMJ) height was recorded using an optoelectric system (Optojump Next, Microgate, SRL, Bolzano, Italy). Three trials were performed, and the best performance was recorded for further analysis, with a rest period of 90 s allowed between trials.

### 2.7. Standing Long Jump

The participants were positioned behind the starting line in a standing position with feet shoulder-width apart and arms loosely hanging down. On the command “ready, set, go”, participants performed a fast downward movement before jumping at maximal effort in the horizontal direction. Participants had to land with both feet at the same time and were not allowed to fall forward, sideward, or backward. The horizontal distance between the starting line and the heel of the rear foot was recorded via a tape measure to the nearest 1 cm. Three trials were performed, and the best performance was recorded for further analysis, with a rest period of 90 s allowed between trials.

### 2.8. Eccentric Training Program

The intervention period lasted eight weeks and took place during the in-season period of the year 2024. Over the course of the eight-week intervention (Table 2), the 2-day group performed combined NHE and RNE training on two days (Tuesdays and Thursdays) per week. Meanwhile, the 1-day group completed the same program on one day per week (Tuesdays). The 1-day group engaged in general skills (submaximal pass-and-move drills) on the day the 2-day group had its second eccentric training session. All training occurred between 17 and 19 o’clock. All groups performed five soccer training sessions per week. The training program was performed prior to team training, on Tuesdays and Thursdays, to ensure it was executed in a non-fatigued state. The NHE was conducted following the protocol used in a previous study of a similar population of youth soccer players [14]. Likewise, the RNE was performed according to the methods outlined in the study by Bouguezzi et al. [16]. Notably, both groups performed an equal volume of eccentric training. Specifically, the total weekly eccentric training duration for both groups was between 10 and 20 min, distributed over either one session (1-day group) or two sessions (2-day group).

## 3. Statistical Analyses

All data analyses were performed using SPSS 25.0 (SPSS, Inc., Chicago, IL, USA). Data are presented as means and standard deviations (SD). Normality assumption was tested and confirmed using the Shapiro–Wilk test. To establish the effect of the interventions on the dependent variables, a 2 (group: 1-day vs. 2-day) × 2 (time: pre vs. post) ANOVA with repeated measures was conducted for each parameter. When group × time interactions reached the significance level (*p* < 0.05), post hoc paired *t*-tests were performed. To determine the magnitude of the training effect, effect sizes (ES) were determined by converting partial eta-squared to Cohen’s d. According to Hopkins et al. [26], ES values were classified as trivial (<0.2), small (0.2–0.6), moderate (0.6–1.2), large (1.2–2.0), very large (2.0–4.0), or extremely large (>4.0). To analyze individual changes, the smallest worthwhile change (SWC_0.2_) was calculated as 0.2 * SD pooled, where SD represents the pooled standard deviation of pre-training scores.

## 4. Results

Participants were rated as pubertal (Table 1). They all received the treatment as allocated, with no training or test-related injuries reported. Physical fitness measures at baseline and follow-up are presented in Table 3. At baseline, no significant between-group differences for anthropometric characteristics or physical fitness emerged (Table 1 and Table 3).

### 4.1. Linear Sprint-Time

The findings indicate a significant main effect of time for the 5 m (ES = 1.83 [Large]) and 10 m (ES = 1.77 [Large]) sprint tests. However, non-significant group × time interactions for the 5 m (ES = 0.34 [small]; *p* > 0.05) and the 10 m (ES = 0.59 [small]; *p* > 0.05) sprint tests were observed (Table 3). Analysis of individual responses demonstrated that the 2-day group had the highest relative number of individual responses > SWC_0.2_ for both the 5 m (77%; *n* = 14) and 10 m sprints (83%; *n* = 15). For the 1-day group, 72% (*n* = 13), and 66% (*n* = 12) of participants displayed improvements > SWC_0.2_.

### 4.2. Change in Direction Speed

Our statistical calculation showed a significant main effect of time (ES = 0.79 [Moderate]) for the 505 CiD speed test. Similarly, a significant group × time interaction (ES = 0.80 [moderate]; *p* < 0.05) emerged (Table 3). The within-group analysis indicated that the 2-day group experienced a moderate pre-to-post improvement (∆7.36%; *p* < 0.001; ES = 0.92). Meanwhile, the 1-day group did not reveal any significant pre-to-post improvement (∆0.05%; *p* > 0.05; ES = 0.00). Whilst 88% of the 2-day group (*n* = 16) demonstrated improvements > SWC_0.2_, only 33% (*n* = 6) of the 1-day group showed improvements > SWC_0.2_.

### 4.3. Y-Agility Test

A large main effect of time (ES = 1.62 [large]) was observed for the Y-agility test. Similarly, a significant group × time interaction was noted (ES = 1.13 [moderate]; *p* < 0.001) (Table 3). Post hoc analyses showed a large performance improvement from pre-to-post for the 2-day group (∆7.91%; ES = 1.68; *p* < 0.001). However, the 1-day group did not reveal any significant pre-to-post improvement (∆1.43%; *p* > 0.05; ES = 0.17). Individual analysis indicated that 100% of the 2-day group (*n* = 18) improved their agility performance > SWC_0.2_, whereas only 44% (*n* = 8) of the 1-day group showed improvement > SWC_0.2_.

### 4.4. Jumping Performance

A significant main effect of time (ES = 0.96 [moderate]) and group × time interaction (ES = 0.83 [moderate], *p* < 0.05) were observed (Table 3). Post hoc analyses demonstrated a small CMJ height improvement for the 2-day group (∆7.44%; *p* < 0.01; ES = 0.35) and no significant pre-to-post change for the 1-day group (∆0.57%; *p* > 0.05; ES = 0.03). In terms of the individual response analysis, the findings show that 66% of the 2-day group (*n* = 11) and only 22% (*n* = 4) of the 1-day group achieved improvements > SWC_0.2_.

Regarding the SLJ, we observed a significant main effect of time (ES = 2.14 [very large]) with no significant group × time interaction (ES = 0.24 [small], *p* > 0.05). Individual response analysis revealed that 88% of the 2-day group (*n* = 16) and 83% (*n* = 15) of the 1-day group reached improvements > SWC_0.2_.

## 5. Discussion

This randomized controlled trial aimed to contrast the effects of volume-equated combined NHE and RNE training performed either once or twice per week on measures of physical fitness in pubertal male soccer players. The main findings indicate that the 2-day eccentric training group showed greater improvements compared to the 1-day group in CiD speed, agility, and CMJ height, with higher proportions of participants exceeding the SWC_0.2_. While both groups similarly improved in sprint and SLJ performance over time, individual response analysis also favored the 2-day group in both tests.

High-intensity actions such as jumping, CiD speed, and agility are key in modern soccer top performance [27]. The results of our study indicate that although the weekly training volume was equal, 2-day eccentric training generated significantly greater benefits than 1-day in CMJ height, CiD speed, and agility. More specifically, the 2-day group displayed significant small-to-large improvements in CMJ height, CiD speed, and agility after 8 weeks of training, while the 1-day group did not show any significant enhancements. These results are further substantiated by the higher proportion of participants who achieved improvements that exceeded the SWC_0.2_ (66 to 100% in the 2-day group, versus 22 to 44% in the 1-day group). Regarding SLJ and linear sprint performance, both groups displayed comparable improvements, but with higher proportions of players who reached enhancements above the SWC_0.2_ in the 2-day group compared to the 1-day group. Taken together, these findings indicate that spreading the same eccentric training volume over 2 sessions per week would be preferable over 1 session in pubertal male soccer players. These results corroborate earlier studies. For example, Moran et al. [15] investigated the effects of an 8-week, volume-equated 1- versus 2-day NHE training on sprinting (10 and 40 m sprint), CiD speed (505 CiD), and jumping (SLJ) in youth male soccer players aged 16 years and revealed that both training formats were effective, with a slight advantage of spreading the volume of training over 2 days instead of 1 day. Additionally, Abdelkader et al. [22] investigated the effects of volume-equated NHE training programs, performed either once or twice per week, on high-intensity actions (30 m sprint, 15 m CiD speed, and SLJ) in prepubertal male soccer players aged 10 years. They revealed significant and overall similar improvements across both training formats for all the tested parameters. However, for the 30 m sprint and SLJ, the 2-day group demonstrated relatively larger enhancements compared with the 1-day group. Taken together with previous research, our findings suggest that, under volume-equated conditions, eccentric exercises such as the NHE and RNE should be distributed across two weekly sessions rather than one. This approach appears to promote more favorable adaptations in key physical fitness components among youth male soccer players.

The improvements in proxies on muscle power, linear sprint speed, CiD speed, and agility performance could be attributed to neural and morphological adaptations [6], although we think that the former contributed to a larger extent compared to the latter, due to the short-duration nature of the training interventions [28,29]. As such, although speculative, the positive adaptations primarily stem from enhanced neural drive to the active muscles [14,15,16]. Nevertheless, morphological adaptation cannot be ruled out. In fact, eccentric training, in particular, is known to increase muscle cross-sectional area more effectively than concentric or traditional training [6,30]. These gains may also be linked to enhanced muscle strength, which would contribute to improved measures of physical fitness. Although these physiological changes were not directly measured, they likely influenced the observed improvements in proxies of muscle power, sprinting, CiD speed, and agility following combined NHE and RNE in pubertal male soccer players.

### Limitations

This study has a number of limitations that warrant discussion. First, although the practical relevance of this study’s findings is not in question, the lack of direct measurements of muscle strength as well as the absence of physiological (muscle activation) and morphological (muscle hypertrophy) assessments constitute limitations of this study. Therefore, future studies should incorporate both performance outcomes and mechanistic assessments to provide a more comprehensive understanding of changes in physical fitness among male youth soccer players following eccentric training. Second, this study was conducted on male youth soccer players, meaning that the outcomes cannot be generalized to female players. Given the generally limited body of studies including female athletes, further studies exploring the effects of eccentric training frequency on physical fitness performance in female players are needed. Third, this study included two experimental groups but lacked an active control group. While the chosen design appropriately addresses the research question, including a control group would have allowed for a clearer comparison of eccentric training-related changes against potential growth-related developments. This limitation should be addressed in future research. Fourth, the duration of the training intervention was relatively short, limiting the ability to project the effects beyond the 8-week intervention period. Therefore, future studies should investigate the impact of longer-term training interventions in youth soccer players to better understand how adaptations to eccentric training evolve over time.

## 6. Conclusions

The main findings of this study suggest that combining NHE and RNE across two sessions per week leads to greater improvements in physical fitness compared to one session per week, under volume-matched conditions. Notably, the 2-day group showed small-to-large improvements in vertical jump, CiD speed, and agility, whereas no significant changes were observed in the 1-day group. Additionally, although both groups demonstrated similar gains in horizontal jump and linear sprint speed, individual response analysis favored the 2-day training frequency. Both the NHE and the RNE are simple to implement and require no equipment, making them highly accessible and cost-effective, particularly for youth soccer clubs with limited facilities. Their practicality allows for easy integration into regular training routines without logistical barriers. Therefore, coaches and practitioners are encouraged to incorporate two weekly sessions of eccentric training involving NHE and RNE in male youth soccer players to support and enhance physical fitness adaptations in a time-efficient and sustainable manner.

## Figures and Tables

**Table 1 jfmk-10-00260-t001:** Anthropometric characteristics of the included participants.

	2-Day Group (*n* = 18)	1-Day Group (*n* = 16)
Age (years)	14.84 ± 0.22	14.58 ± 0.28
Body height (cm)	171.11 ± 7.32	171.29 ± 6.33
Body mass (kg)	58.25 ± 6.52	58.44 ± 9.39
Maturity offset (years) *	1.17 ± 0.40	1.02 ± 0.25
APHV	13.60 ± 0.39	13.56 ± 0.42

Notes: Data are presented as means and standard deviations; *: as years from peak height velocity. APHV = age at peak height velocity.

**Table 2 jfmk-10-00260-t002:** Eccentric training program.

2-Day Group					1-Day Group		
Week	Exercise	Sets	Repetitions/Sets	Rest Between Sets	Sets	Repetitions/Sets	Rest Between Sets
1	NHE	2	6 to 8	60 to 90	4	6 to 8	60 to 90
	RNHE	2	6 to 8	60 to 90	4	6 to 8	60 to 90
2	NHE	2	8	60 to 90	4	8	60 to 90
	RNHE	2	8	60 to 90	4	8	60 to 90
3	NHE	4	6	60 to 90	8	6	60 to 90
	RNHE	4	6	60 to 90	8	6	60 to 90
4	NHE	4	8	60 to 90	8	8	60 to 90
	RNHE	4	8	60 to 90	8	8	60 to 90
5	NHE	3	8	60 to 90	6	8	60 to 90
	RNHE	3	8	60 to 90	6	8	60 to 90
6	NHE	4	10	60 to 90	8	10	60 to 90
	RNHE	4	10	60 to 90	8	10	60 to 90
7	NHE	4	12	60 to 90	8	12	60 to 90
	RNHE	4	12	60 to 90	8	12	60 to 90
8	NHE	4	12	60 to 90	8	12	60 to 90
	RNHE	4	12	60 to 90	8	12	60 to 90

NHE: Nordic hamstring exercise; RNHE: reverse Nordic hamstring exercise.

**Table 3 jfmk-10-00260-t003:** Group-specific changes in measures of physical fitness from pre-to-post training.

	2-Day Group (*n* = 18)				1-Day Group (*n* = 16)				ANOVA	
	Pretest		Posttest		Pretest		Posttest		*p*-Value (ES)	
	M	SD	M	SD	M	SD	M	SD	Time	Group × Time
Linear sprint speed										
5 m sprint (s)	1.15	0.06	1.10	0.06	1.15	0.05	1.11	0.04	<0.001 (1.83)	>0.05 (0.34)
10 m sprint (s)	1.95	0.11	1.89	0.08	2.00	0.10	1.97	0.08	<0.01 (1.77)	>0.05 (0.59)
Change in direction speed										
505 CiD speed (s)	2.54	0.11	2.45	0.09	2.67	0.17	2.67	0.13	<0.05 (0.79)	<0.05 (0.80)
Agility										
Y-shaped agility (s)	2.80	0.14	2.58	0.13	2.82	0.25	2.72	0.23	<0.001 (1.62)	<0.01 (1.13)
Proxies of Muscle Power										
CMJ (cm)	27.54	5.57	29.59	6.61	26.52	5.05	26.67	5.35	<0.05 (0.96)	<0.05 (0.83)
SLJ (m)	2.07	0.26	2.21	0.24	2.00	0.17	2.11	0.21	<0.001 (2.14)	>0.05 (0.24)

M: mean; SD: standard deviation; SLJ: standing long jump; CMJ: countermovement jump; CiD: change in direction; ES: effect size.

## Data Availability

The original data presented in the study are openly available in FigShare at 10.6084/m9.figshare.29178872.

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
