# Peer review of "The Effects of Volume-Matched One-Day Versus Two-Day Eccentric Training on Physical Performance in Male Youth Soccer Players"

_jfmk, 2025, doi:10.3390/jfmk10030260_

Round 1
Reviewer 1 Report
Comments and Suggestions for Authors
Thank you for permitting me to review this manuscript
In this study the effect of one day vs two days eccentric training was assessed on youth soccezr players , they found the two daus trainig was superior in comparison to one day trainig
here are my comments
abstract
please add also the negative findings
Introduction
Line 56 please provide reference (PPR)
methods:
Please describe exactly the total time of eccentric training for each group , as I understanf it is the same but it is not very clear in the text
stattistical analysis
I suggest sample size analysis should be transferrred from line 100 to this section
results:
linear sprint time, change of direction speed and agility test was the analysis of individual response significantly different , if not it could may be it should be better deleted
Discussion
the aim of the study should not be recited n and the result may be described
limitations , plesase developp better the future dstudies should incrorporated :
Author Response
Reviewer 1
Thank you for permitting me to review this manuscript
In this study the effect of one day vs two days eccentric training was assessed on youth soccezr players , they found the two daus trainig was superior in comparison to one day trainig
here are my comments
abstract
please add also the negative findings
Reply: Thank you for your comment. We believe that the results are transparently reported. More specifically, we reported either the significant and non-significant group-by-time interaction as follows “Significant group-by-time interactions were observed for the 505 CoD test, agility, and CMJ performance (effect sizes [ES] = 0.80 to 1.13). However, no significant interactions were found for the 5 and 10 m sprint as well as SLJ (p > 0.05).” Similarly, negative findings were further highlighted while reporting the within-group pre-post differences as follows “Compared to the 1-day group, the 2-day training group showed greater improvements in CoD speed (∆7.36%; p < 0.001; ES = 0.92), agility (∆7.91%; p < 0.001; ES = 1.68), and CMJ (∆7.44%; p < 0.01; ES = 0.35), while no differences were observed in 5, and 10 m linear sprint or SLJ performance.”
Introduction
Line 56 please provide reference (PPR)
methods:
Please describe exactly the total time of eccentric training for each group , as I understand it is the same but it is not very clear in the text
Reply: The total training volume was matched between the two training formats, ensuring that the only variable difference was the training frequency. This approach was essential to maintain high internal validity of the findings. In line with the reviewer’s suggestion, the exact duration of the training has been added to the revised version as follows:" Notably, both groups performed an equal volume of eccentric training. Specifically, the total weekly eccentric training duration for both groups was between 10 to 20 minutes, distributed over either one session (1-day group) or two sessions (2-day group).”
statistical analysis
I suggest sample size analysis should be transferred from line 100 to this section
Reply: Thank you for your suggestion. We believe that the a priori power analysis is best placed in the “Participants” section. This section details the number of participants, their characteristics, and inclusion criteria, making it the most logical and informative location for justifying the sample size to the reader. Including the power analysis here ensures transparency regarding how the sample size was determined in relation to the study population. While some journals may report it in the “Statistical Analysis” section, placing it in the “Participants” section is common practice in many peer-reviewed publications and aligns well with standard reporting guidelines.
results:
linear sprint time, change of direction speed and agility test was the analysis of individual response significantly different , if not it could may be it should be better deleted
Reply: Thank you for your comment. The individual response analysis was conducted as a supplementary approach to offer additional insight beyond group-level comparisons by focusing on individual variability. This perspective provides a clearer understanding of how participants responded to each training program on an individual level. While group differences were statistically analyzed using mean values, the individual response analysis was descriptive and not subjected to statistical comparison. We reported the percentage and number of participants in each group who exceeded the SWC0.2 threshold, allowing interpretation of which group showed a greater proportion of meaningful individual responses.
Discussion
the aim of the study should not be recited n and the result may be described
Reply: Thank you for your suggestion. Reiterating the main purpose of the study at the beginning of the discussion section is a common and well-accepted practice. We find it particularly helpful in keeping the reader focused on the study’s key objectives before diving into the interpretation of the findings. The main results are briefly summarized to provide context for the more detailed discussion that follows. We have made minor adjustments to this section in the revised version to enhance clarity.
limitations , plesase developp better the future dstudies should incrorporated :
Reply: Thank you for your suggestion. We expanded the limitation section per the reviewer’s suggestion as follows:
“This study has a number of limitations that warrant discussion. First, although the practical relevance of this study’s findings is not in question, the lack of direct measurements of muscle strength as well as the absence of physiological (muscle activation) and morphological (muscle hypertrophy) assessments constitute limitations of this study. Therefore, future studies should incorporate both performance outcomes and mechanistic assessments to provide a more comprehensive understanding of changes in physical fitness among male youth soccer players following eccentric training. Second, this study was conducted on male youth soccer players, meaning that the outcomes cannot be generalized to female players. Given the generally limited body of studies including female athletes, further studies exploring the effects of eccentric training frequency on physical fitness performance in female players are needed. Third, this study included two experimental groups but lacked an active control group. While the chosen design appropriately addresses the research question, including a control group would have allowed for a clearer comparison of eccentric training-related changes against potential growth-related developments. This limitation should be addressed in future research. Fourth, the duration of the training intervention was relatively short, limiting the ability to project the effects beyond the 8-week intervention period. Therefore, future studies should investigate the impact of longer-term training interventions in youth soccer players to better understand how adaptations to eccentric training evolve over time.”
Reviewer 2 Report
Comments and Suggestions for Authors
Dear authors,
Congratulations on your excellent study, as well as for the clarity with which it is presented in this manuscript. Below you will find an evaluation of your manuscript section by section.
ABSTRACT: The abstract is clear, well-structured, and comprehensive. It concisely presents the aim, study design, main results (including effect sizes), and the practical conclusion.
INTRODUCTION: A strong and well-documented introduction that clearly outlines the context, research gaps, and focused aim of the study.
METHODS: The Methods section is extensive, carefully organized, and fully informative, covering all the essential components of a properly documented randomized controlled trial (RCT). However: (a) the method of randomization is not described (e.g., random number generator, sealed envelopes). Including this information would improve transparency; (b) in lines 106–107, the 1-day group is reported as (n=18). Is this correct? In other parts, this group is described as n=16. Please clarify which value is accurate.
RESULTS: The Results section is excellently structured, very clear, and easy to read. It begins with key information on protocol adherence, absence of injuries, and lack of baseline differences (all important for the internal validity of the study). It then proceeds with four subsections based on each test, which enhances readability.
DISCUSSION: The Discussion is structured, clear, and focused. It begins with the key findings (in relation to the study’s aim) and expands into interpretation based on the international literature and physiological rationale. It also includes the main limitations of the study, reinforcing its scientific integrity. However, it could benefit from a more detailed practical recommendation for implementation in training routines, as well as suggestions for future research.
CONCLUSIONS: The Conclusions section is concise, clear, and fully aligned with the study’s data.
Author Response
Reviewer 2
Dear authors,
Congratulations on your excellent study, as well as for the clarity with which it is presented in this manuscript. Below you will find an evaluation of your manuscript section by section.
Reply: Thank you for your affirmative comment. Highly appreciated.
ABSTRACT: The abstract is clear, well-structured, and comprehensive. It concisely presents the aim, study design, main results (including effect sizes), and the practical conclusion.
Reply: Thank you for your positive feedback
INTRODUCTION: A strong and well-documented introduction that clearly outlines the context, research gaps, and focused aim of the study.
Reply: Thank you for your positive feedback
METHODS: The Methods section is extensive, carefully organized, and fully informative, covering all the essential components of a properly documented randomized controlled trial (RCT). However: (a) the method of randomization is not described (e.g., random number generator, sealed envelopes). Including this information would improve transparency; (b) in lines 106–107, the 1-day group is reported as (n=18). Is this correct? In other parts, this group is described as n=16. Please clarify which value is accurate.
Reply: Thank you for your positive feedback. Further details pertaining to the randomization process were added as follows: “Participants were randomly assigned to a two-day group (n=18) or a 1-day group (n=16) using sealed, opaque envelopes to ensure allocation concealment. The anthropometric characteristics of both groups are displayed in Table 1.”
Thank you for bringing this mistake to our attention. The correct number is 16. This is now fixed in the revised version.
RESULTS: The Results section is excellently structured, very clear, and easy to read. It begins with key information on protocol adherence, absence of injuries, and lack of baseline differences (all important for the internal validity of the study). It then proceeds with four subsections based on each test, which enhances readability.
Reply: Thank you for your positive feedback.
DISCUSSION: The Discussion is structured, clear, and focused. It begins with the key findings (in relation to the study’s aim) and expands into interpretation based on the international literature and physiological rationale. It also includes the main limitations of the study, reinforcing its scientific integrity. However, it could benefit from a more detailed practical recommendation for implementation in training routines, as well as suggestions for future research.
Reply: Thank you for your positive feedback. According to the reviewer’s suggestion, the practical applications section has been expanded as follows: “The main findings of this study suggest that combining NHE and RNE across two sessions per week leads to greater improvements in physical fitness compared to one session per week, under volume-matched conditions. Notably, the 2-day group showed small-to-large improvements in vertical jump, CoD speed, and agility, whereas no significant changes were observed in the 1-day group. Additionally, although both groups demonstrated similar gains in horizontal jump and sprint speed, individual response analysis favored the 2-day training frequency. Both the NHE and the RNE are simple to implement and require no equipment, making them highly accessible and cost-effective, particularly for youth soccer clubs with limited facilities. Their practicality allows for easy integration into regular training routines without logistical barriers. Therefore, coaches and practitioners are encouraged to incorporate two weekly sessions of eccentric training involving NHE and RNE in youth male soccer players to support and enhance physical fitness adaptations in a time-efficient and sustainable manner.”
In addition, more details related to the potential future investigations were added as follows: “This study has a number of limitations that warrant discussion. First, although the practical relevance of this study’s findings is not in question, the lack of direct measurements of muscle strength as well as the absence of physiological (muscle activation) and morphological (muscle hypertrophy) assessments constitute limitations of this study. Therefore, future studies should incorporate both performance outcomes and mechanistic assessments to provide a more comprehensive understanding of changes in physical fitness among male youth soccer players following eccentric training. Second, this study was conducted on male youth soccer players, meaning that the outcomes cannot be generalized to female players. Given the generally limited body of studies including female athletes, further studies exploring the effects of eccentric training frequency on physical fitness performance in female players are needed. Third, this study included two experimental groups but lacked an active control group. While the chosen design appropriately addresses the research question, including a control group would have allowed for a clearer comparison of eccentric training-related changes against potential growth-related developments. This limitation should be addressed in future research. Fourth, the duration of the training intervention was relatively short, limiting the ability to project the effects beyond the _8-week intervention period. Therefore, future studies should investigate the impact of longer-term training interventions in youth soccer players to better understand how adaptations to eccentric training evolve over time.”
CONCLUSIONS: The Conclusions section is concise, clear, and fully aligned with the study’s data.
Reply: Thank you for your positive feedback.